# Biologically plausible solutions for spiking networks with efficient coding

**Veronika Koren**
Department of Excellence for Neural Information Processing
Center for Molecular Neurobiology (ZMNH)
University Medical Center Hamburg-Eppendorf (UKE)
Falkenried 94, 20251 Hamburg, Germany
v.koren@uke.de

**Stefano Panzeri**
Department of Excellence for Neural Information Processing
Center for Molecular Neurobiology (ZMNH)
University Medical Center Hamburg-Eppendorf (UKE)
Falkenried 94, 20251 Hamburg, Germany
Istituto Italiano di Tecnologia
Genoa, Italy
s.panzeri@uke.de

## Abstract

Understanding how the dynamics of neural networks is shaped by the computations they perform is a fundamental question in neuroscience. Recently, the framework of efficient coding proposed a theory of how spiking neural networks can compute low-dimensional stimulus signals with high efficiency. Efficient spiking networks are based on time-dependent minimization of a loss function related to information coding with spikes. To inform the understanding of the function and dynamics of biological networks in the brain, however, the mathematical models have to be informed by biology and obey the same constraints as biological networks. Currently, spiking network models of efficient coding have been extended to include some features of biological plausibility, such as architectures with excitatory and inhibitory neurons. However, biological realism of efficient coding theories is still limited to simple cases and does not include single neuron and network properties that are known to be key in biological circuits. Here, we revisit the theory of efficient coding with spikes to develop spiking neural networks that are closer to biological circuits. Namely, we find a biologically plausible spiking model realizing efficient coding in the case of a generalized leaky integrate-and-fire network with excitatory and inhibitory units, equipped with fast and slow synaptic currents, local homeostatic currents such as spike-triggered adaptation, hyperpolarization-activated rebound current, heterogeneous firing thresholds and resets, heterogeneous postsynaptic potentials, and structured, low-rank connectivity. We show how the complexity of E-E connectivity matrix shapes network responses.

## 1 Introduction

Complex biological systems are thought to be shaped through evolution to perform optimally a set of desired functions within the constraints of their biological features. A bulk of previous influential work [34, 25, 9] has proposed that the organization of the mammalian cortex, the most sophisticated

biological organ, makes no exception and follows too the general principles of evolution towards optimality under biological constraints.

One prominent theory is that the function of the cerebral cortex is organized for efficient coding, in that it optimizes the encoding of information under a number of biological constraints such as the metabolic expenditure, the speed and the timing of information coding. Imposed constraints can be formalized with a loss function.

To understand biological neural networks that operate in animal's brains, the mathematical description of networks optimizing functionally relevant loss functions has to include as many features of biological realism as possible, so that the constraints satisfied by the model network will be informative of those obeyed by biological neural circuits. Thus, all elements of the model should have a counterpart in biological circuits [15] and be directly relatable to them. It is still unclear how to derive efficient coding theories of biologically plausible neuronal networks.

An influential proposal of efficient coding in cortical circuits has been that visual cortical circuits extract statistical regularities from static natural images with a sparse set of basis functions, which yielded basis functions that resemble receptive fields in primary visual cortex (V1) [34, 35]. A related proposal in the domain of auditory processing yielded auditory circuits that extract statistical regularities of natural sounds with a set of basis functions that resemble the tuning properties of auditory nerve fibers [25]. In these studies, the goal of efficient coding is to learn sparse spatial or temporal filters that enables optimal information encoding about the stimulus by the activity of each single neuron. Analyses of cortical networks, however, show that neurons are recurrently connected and that information processing tends to be distributed across neurons [18]. In recurrently connected circuits, neuron-to-neuron interactions typically strongly impact on the information encoded in the spiking activity [43] as well as on how this information is transmitted to and read out by downstream structures to produce relevant behavioral outputs [44, 48, 37]. Moreover, analyses of spike trains recorded in the cerebral cortex show that millisecond precision of spike timing of sensory neurons carries information about both static [40, 36] and dynamic stimuli [20, 32], which suggests that the temporal information in spiking activity is highly relevant for the information processing in the cortex. Thus, analyses of empirical neural data strongly suggest that biologically relevant theories of coding in the cerebral cortex must rely on spiking neuron models and on population codes.

To address this issue, models of how to optimally perform a given coding function have been extended to spiking dynamics [3]. Models of efficient coding with spikes are developed from a loss function that includes a quadratic error between the desired signal and its estimate (see the Supplementary material). Equations for the subthreshold dynamics of membrane potentials and firing thresholds are analytically derived by assuming that the error is minimized with every spike [1, 5]. However, the efficient coding with spikes, and the related theory of predictive coding [51], have been so far developed as abstract mathematical frameworks that lack biological realism. Namely, these models are not consistent with the observation that a single neuron can be either excitatory or inhibitory, but not both (Dale's principle, [11, 19]). Very recently, progress has been made in this direction by deriving efficient coding with spiking networks with a more biologically plausible neural circuit architecture with interacting excitatory (E) and inhibitory (I) neurons [26], opening the possibility to model neural computations with biologically relevant models.

Here, we make major further progress by deriving a mathematically well-defined extension of efficient coding for considerably more biologically plausible spiking neural network models of cortical circuits. We develop a recurrently connected neural circuit with E and I neurons that performs a generic linear transformation between its inputs and its outputs. We impose the E-I architecture and recover a network with low-rank and structured connectivity that maximizes the information transmission about the stimulus with every spike. Here, we derive a solution of the spiking network model from optimality principles that includes many desirable and biologically realistic properties. Our solution takes the form of a biologically plausible generalized leaky integrate-and-fire (LIF) neuron model and satisfies the principle of functional specificity of excitatory and inhibitory neurons (Dale's principle). Our solution also prescribes faster membrane time constant of I compared to E neurons, compatible with the biophysical properties of pyramidal neurons and fast-spiking inhibitory interneurons. Our derivations prescribe subthreshold dynamics, firing thresholds and resets after spiking, thus giving a complete description of a spiking neural network along with network's computation. Importantly, all elements of the model are biologically plausible and directly relatable to empirically measurable currents in neurons.

The paper is structured as follows. We first introduce the loss functions and explaining how the spike timing is conditioned on a minimization of loss functions of E and I neurons in every time step. We then outline the analytical treatment of loss functions that leads to expressions of membrane potentials and firing thresholds of a generalized LIF network model with structured, low-rank connectivity. Further, we provide a description and biologically plausible interpretation of derived currents. Finally, we simulate the network with biologically plausible parameters and analyze the behavior of the network with respect to the complexity of E-E synaptic connections.

## 2 Results

### 2.1 Minimization of the loss function with spiking neurons

From a spiking network of $N^E$ excitatory and $N^I$ inhibitory neurons, we define a linear population readout of the spiking activity of E and I neurons:

$$\dot{\hat{\boldsymbol{x}}}_y(t) = -\frac{1}{\tau_y}\hat{\boldsymbol{x}}_y(t) + W_Y \boldsymbol{f}_y(t), \qquad y \in \{E, I\}, \tag{1}$$

with $\tau_y > 0$ the time constant of the population readout. In the population readout, the decoding matrix $W_y$ weights the vector of spikes $\boldsymbol{f}_y(t) = [f_1^y(t), \ldots, f_{N_y}^y(t)]^\intercal$. The $M$-by-$N_y$ matrix $W_y$ associates each of the $N_E$ excitatory ($N_I$ inhibitory) neurons with $M$ weights, $W_y = (w_{mi}^y); m = 1 \ldots, M, \ i = 1, \ldots, N_y$, where $N_y$ is the number of neurons of the cell type $y$, and $M$ is the number of stimulus features estimated by the network. The population readout in eq. (1) is an estimate of the desired signal $\boldsymbol{x}(t)$, that is not directly accessible to the network.

We assume that the desired signal $\boldsymbol{x}(t)$ is a linear function of stimulus features $\boldsymbol{s}(t) = [s_1(t), \ldots, s_m(t), \ldots, s_M(t)]^\intercal$ and is defined as follows:

$$\dot{\boldsymbol{x}}(t) = A\boldsymbol{x}(t) + \boldsymbol{s}(t), \tag{2}$$

where $A = (a_{mn}); \ m, n = 1, ..., M$ is a square matrix determining the linear transformation between the input features $\boldsymbol{s}(t)$ and the desired signal $\boldsymbol{x}(t)$. We will refer to the transformation between the features and the desired signal as network's computation. If $A$ is a diagonal matrix, network's computation operates independently across features, while a non-diagonal matrix $A$ implements linear mixing of features.

We hypothesize that the objective of the excitatory network is to minimize the distance between the desired signal and the population readout of E neurons, while the objective of the inhibitory population is to minimize the distance between the readout of the E and I populations [3, 7]. Besides the coding error, we also take into account the metabolic cost on spiking, since spiking activity in the brain is energetically expensive [33]. These objectives are formalized by two loss functions with a quadratic coding error and a quadratic regularizer [16], relative to E and I cell type:

$$\mathrm{L}_E(t) = \sum_{m=1}^{M} (x_m(t) - \hat{x}_m^E(t))^2 + \mu_E \sum_{i=1}^{N_E} (r_i^E(t))^2 \tag{3a}$$

$$\mathrm{L}_I(t) = \sum_{m=1}^{M} (\hat{x}_m^E(t) - \hat{x}_m^I(t))^2 + \mu_I \sum_{i=1}^{N_I} (r_i^I(t))^2, \tag{3b}$$

where constants $\mu_E, \mu_I > 0$ are weighting the relative contribution of the metabolic cost over the coding error. The low-pass filtered spike train of the neuron $i$ is defined as

$$\dot{r}_i^y = -\frac{1}{\tau_i^{r,y}} r_i^y(t) + f_i^y(t), \qquad y \in \{E, I\}, \tag{4}$$

with $\tau_i^{r,y} > 0$ the time constant of the low-pass filter. Gathering the variables $r_i^y(t)$ across neurons, we define a vector of time-dependent neural activities $\boldsymbol{r}_y(t) = [r_1^y(t), \ldots, r_{N_y}^y(t)]^\intercal$. Note that an alternative to the loss function in eq. (3b) would be to minimize the distance between the readout of the I population and the signal. However, such a formulation, without further approximations, leads to a network where E and I neurons are unconnected.

We assume that neuron $i$ of cell type $y$ will fire a spike at time $t = t^+$ if and only if this decreases the loss function, i.e.

$$\mathrm{L}_y \left( t^+ \,|\, \left[ f_i^y(t^+) = 1 \right] + \eta_i^y(t^+) \right) < \mathrm{L}_y \left( t^- \,|\, \left[ f_i^y(t^-) = 0 \right] \right), \tag{5}$$

with the noise term $\eta_i^y(t) = \sigma_i^y \xi_i^y(t)$. Gaussian random variable $\xi_i^y(t)$ has zero mean and covariance $\langle \xi_i(t) \xi_j(t') \rangle = \delta_{ij} \delta(t - t')$, and $\sigma_i^y$ is the noise intensity. Contrary to previous approaches where the noise is added in the membrane potential [3, 23], we here assume noise in the condition for spiking (eq. 5). While the noise in the membrane potential models synaptic inputs that are unrelated to the coding function of the network, the noise as in eq. 5 captures the noise in spike generation for a given membrane potential, a source of noise that characterizes spike generation in biological neurons [12]. In biological neurons, a spike is initiated when a sufficient fraction of sodium channels in the neural membrane is opened, and the fraction of open channels for a given membrane potential is probabilistic.

Assuming that a spike in neuron $i$ at time $t^+$ is only fired if this decreases the loss function in eq. (5), we apply an analytical treatment of eqs. 3a-3b similar to [3, 23], and obtain the following expressions:

$$\begin{aligned} \boldsymbol{w}_E^\mathsf{T} \left( \boldsymbol{x}(t) - \hat{\boldsymbol{x}}_E(t) \right) - \mu_E r_i^E(t) &> \frac{1}{2} \left( \| \boldsymbol{w}_i^E \|_2^2 + \mu_E \right) + \sigma_i^E \xi_i^E(t) \\ \boldsymbol{w}_I^\mathsf{T} \left( \hat{\boldsymbol{x}}_E(t) - \hat{\boldsymbol{x}}_I(t) \right) - \mu_I r_i^I(t) &> \frac{1}{2} \left( \| \boldsymbol{w}_i^I \|_2^2 + \mu_I \right) + \sigma_i^I \xi_i^I(t) \end{aligned} \tag{6}$$

with $\| \boldsymbol{w}_i^y \|_2^2$ the squared length of decoding vector of the neuron $i$, $\| \boldsymbol{w}_i^y \|_2^2 = \sum_{m=1}^M (w_{mi}^y)^2$. Since we assumed threshold crossing, we interpret the left-hand side of eq. (6) as the membrane potential and the right-hand side as the firing threshold of the neuron $i$. Note that the firing threshold has a deterministic and a stochastic part. Collecting membrane potentials across neurons, $\boldsymbol{u}_y(t) = [u_1^y(t), \ldots, u_{N_y}^y(t)]^\mathsf{T}$ with $y \in \{E, I\}$, we express them with vector notation as follows:

$$\begin{aligned} \boldsymbol{u}_E(t) &\equiv W_E^\mathsf{T} \left( \boldsymbol{x}(t) - \hat{\boldsymbol{x}}_E(t) \right) - \mu_E \boldsymbol{r}_E(t) \\ \boldsymbol{u}_I(t) &\equiv W_I^\mathsf{T} \left( \hat{\boldsymbol{x}}_E(t) - \hat{\boldsymbol{x}}_I(t) \right) - \mu_I \boldsymbol{r}_I(t). \end{aligned} \tag{7}$$

To express the temporal dynamics of the membrane potentials, $\dot{\boldsymbol{u}}_E(t)$ and $\dot{\boldsymbol{u}}_I(t)$, we take the time-derivative of all time-dependent terms in eq. (6). Contrary to previous approaches and without loss of generality, we express the mixing matrix $A$ as a difference of a square matrix $B$ and a diagonal matrix $\lambda_E \mathbf{I}^{MxM}$,

$$A = B - \lambda_E \mathbf{I}^{MxM}, \tag{8}$$

with $\mathbf{I}$ the identity matrix. Seen that loss functions minimize the distance between the signal and the E estimate (eq. 3a), and between the E and the I estimates (eq. 3b), we use the approximations $\boldsymbol{x}(t) \approx \hat{\boldsymbol{x}}_E(t)$ and $\hat{\boldsymbol{x}}_E(t) \approx \hat{\boldsymbol{x}}_I(t)$. Moreover, we remove synaptic connections that violate Dale's law (see the Supplementary material). After re-arranging the terms and without further approximations, interestingly, the network model can be expressed as a generalized LIF model with structured connectivity.

## 2.2 Generalized integrate-and-fire network with low-rank structured connectivity

Due to substitutions of the signal and the estimates, we write the membrane potential as $u_i^y(t) \approx V_i^y(t)$. The subthreshold dynamics of E and I neurons is then dependent on the following currents:

$$\tau_E \dot{V}_i^E(t) = -V_i^E(t) + I_i^{\mathrm{ff}}(t) + I_i^{EE}(t) + I_i^{EI}(t) + I_i^{\mathrm{local}\ E}(t) + I_i^{\mathrm{rebound}}(t), \tag{9a}$$

$$\tau_I \dot{V}_i^I(t) = -V_i^E(t) + I_i^{IE}(t) + I_i^{II}(t) + I_i^{\mathrm{local}\ I}(t), \tag{9b}$$

and complemented with fire-and-reset rule: if $V_i^y(t^-) \geq \vartheta_i^y(t^-) \rightarrow V_i^y(t^+) = V_i^{\mathrm{reset}\ y}$. For simplicity, we set the resting potential of all neurons to $V^{\mathrm{rest}} = 0$, but a biologically plausible resting potential can be introduced in eqs. (9a)-(9b) without affecting network's dynamics. From eq.(6), the firing threshold that takes into account the noise in spike generation (eq. 5) is the following:

$$\vartheta_i^y(t) = \frac{1}{2}(\mu_y + \| \boldsymbol{w}_i^y \|_2^2) + \sigma_i^y \xi_i^y(t), \qquad y \in \{E, I\}, \tag{10}$$

while the reset potentials for E and I cell types are

$$V_i^{\text{reset } E} = -\frac{1}{2}(\mu_E - \|\boldsymbol{w}_i^E\|_2^2)$$

$$V_i^{\text{reset } I} = -\frac{1}{2}(\mu_I + \|\boldsymbol{w}_i^I\|_2^2). \tag{11}$$

The firing thresholds in eq. 10 and reset potentials in eq. 11 are thus proportional to the length of decoding vector of the local neuron $\|\boldsymbol{w}_i^y\|_2$, and to the regularizer $\mu_y$ that affects equally all the neurons of the same cell type. Note that if decoding weights $w_{mi}^y$ are heterogeneous across neurons, firing thresholds and resets are heterogeneous as well.

The membrane equations in eqs. 9a-9b depend on a number of currents that, interestingly, all have a straightforward counterpart in biological networks. The first currents on the right-hand site of eqs. 9a-9b are leak currents, that result from absorbing of several terms, among others the diagonal matrix $\lambda_E \mathbf{I}^{MxM}$ from eq. 8. The feedforward current to E neurons is given by a weighted sum of external inputs $s_k(t)$,

$$I_i^{\text{ff}}(t) = \tau_E \sum_{m=1}^{M} w_{mi}^E s_m(t). \tag{12}$$

Inhibitory synaptic currents to the postsynaptic neuron $i$ are given by a weighted sum of spikes of presynaptic neurons,

$$I_i^{II}(t) = -\tau_I \sum_{\substack{j=1 \\ j \neq i}}^{N_I} C_{ij}^{II} f_j^I(t), \qquad I_i^{EI}(t) = -\tau_I \sum_{j=1}^{N_I} C_{ij}^{EI} f_j^I(t), \tag{13}$$

with structured connectivity matrices $C_{ij}^{yz}$, where the strength of synaptic connection is proportional to the similarity of decoding vectors of the presynaptic neuron $j$ and the postsynaptic neuron $i$,

$$C_{ij}^{yz} = \begin{cases} (\boldsymbol{w}_i^y)^\intercal \boldsymbol{w}_j^z, & \text{if } (\boldsymbol{w}_i^y)^\intercal \boldsymbol{w}_j^z > 0 \\ 0, & \text{otherwise.} \end{cases} \tag{14}$$

Note that synaptic connections between neurons with different selectivity (i.e., neuronal pairs with negative dot product of decoding vectors, $(\boldsymbol{w}_i^y)^\intercal \boldsymbol{w}_j^z < 0$), have been set to zero to make the network consistent with Dale's law (see the Supplementary material). In particular, we had to ensure that a particular neuron can only send excitatory or inhibitory currents to other neurons, but not both. Even though there is no direct fast synaptic connections between E-to-E neurons, effectively, fast synaptic connectivity implements lateral inhibition between E neurons with similar selectivity. Lateral inhibition (or competition) between E neurons with similar selectivity is a dynamical effect that is essential for an efficient neural code [21] and has been demonstrated in biological circuits [8].

While local inhibitory synapses have fast kinetics of spike trains (eq. 13), recurrent excitatory synaptic currents are slower since they are convolved with the synaptic filter $z_j^E(t)$:

$$I_i^{EE}(t) = \tau_E \sum_{\substack{j=1 \\ j \neq i}}^{N_E} D_{ij}^{EE} z_j^E(t)$$

$$I_i^{IE}(t) = \tau_E \sum_{j=1}^{N_E} C_{ij}^{IE} f_j^E(t) + \left(\frac{\tau_E}{\tau_I} - 1\right) \sum_{j=1}^{N_E} D_{ij}^{IE} z_j^E(t), \qquad \tau_E > \tau_I \tag{15}$$

$$\dot{z}_j^E(t) = -\frac{1}{\tau_E^{\text{syn}}} z_j^E(t) + f_j^E(t), \qquad \tau_E^{\text{syn}} = \tau_E,$$

with matrices of synaptic interactions:

$$D_{ij}^{yE} = \begin{cases} (\boldsymbol{w}_i^y)^\intercal B \boldsymbol{w}_j^E, & \text{if } (\boldsymbol{w}_i^y)^\intercal \boldsymbol{w}_j^E > 0, \qquad B \text{ positive semi-def.,} \\ 0, & \text{otherwise.} \end{cases} \tag{16}$$

where $y = E$ for E-to-E synapses and $y = I$ for E-to-I synapses. Same as with fast synaptic currents in eq. (13), the strength of the synapse between the presynaptic neuron $j$ and the postsynaptic neuron

$i$ in eq. (16) depends on the similarity of decoding vectors of the presynaptic and postsynaptic neurons, $(\boldsymbol{w}_i^y)$ and $\boldsymbol{w}_j^E$. To make slower synaptic currents consistent with Dale's law, we removed connections between neurons with different selectivity, i.e., connections for which the following is true: $(\boldsymbol{w}_i^y)^\intercal \boldsymbol{w}_j^z < 0$ (see eq. 16). In addition, slower synaptic currents depend on the matrix $B$ that was expressed from the mixing matrix $A$ (see eq. 8). To ensure that presynaptic excitatory neurons always cause an excitatory current in the postsynaptic neuron, we constrained the matrix $B$ to positive semi-definite. Moreover, the excitatory effect of an E-to-I synapse constrains the following relation of time constants: $\tau_I < \tau_E$. Since $\tau_E$ and $\tau_I$ are membrane time constants of the E and I cell type, respectively, this relation constrains the membrane time constant in E neurons to be slower than in I neurons, consistent with E cell type modeling pyramidal cells while the I cell type models fast-spiking interneurons [30]. Fast-spiking (somatostatin) interneurons are a prevalent class of inhibitory neurons in cortical circuits [47]. As pyramidal neurons account for about 80 % of cortical neurons, and fast-spiking interneurons account for at least a half of inhibitory neurons in the cortex [49], our model altogether provides a description for about 90 % of cortical neurons. Note that alternative solutions for slow currents can be formulated (see the Supplementary Material). These alternative solutions are, however, of lesser biological relevance since they either result in a network without E-to-E connections, or lead to a global imbalance of E and I currents in the network. Contrary to fast connections that are expected to implement lateral inhibition among E neurons with similar selectivity (see eq. 13), slower E-to-E synaptic connections are expected to implement cooperation among excitatory neurons with similar selectivity.

The efficient E-I network has low-rank connectivity. The rank of fast synaptic connectivity is, by definition of the connectivity matrices $C$ (eq. 14) and $D$ (eq. 16), equivalent to the maximal number of features encoded by the network, $M$. We assume that the number of active features $s_m(t)$ is much smaller than the number of neurons in the network, $M << N^E$, which gives low-rank connectivity matrices [29]. Low-rank connectivity constrains the neural activity to a low-dimensional manifold, and the relevance of the latter for the description of the dynamics of biological neural ensembles has been strongly suggested by empirical neural recordings [13, 50]. Note also that using heterogeneous and sparse decoding weights $w_i^y$ gives heterogeneous and sparse synaptic connectivity.

Next, we have currents that are triggered by spiking of the local neuron,

$$I_i^{\text{local } y}(t) = -\mu_y \left(1 - \frac{\tau_y}{\tau_i^{r,y}}\right) r_i^y(t), \qquad y \in \{E, I\}, \tag{17}$$

with $r_i^y(t)$ the single neuron readout with time constant $\tau_i^{r,y}$ (eq. 4). While the kinetics of the local current is given by the single neurons readout, the sign and the strength of the current depend on the relation of time constants between the population readout $\tau_y$, and the single neuron readout $\tau_i^{r,y}$. To allow the population readout $\hat{\boldsymbol{x}}_y(t)$ to track fast changes of the signal $\boldsymbol{x}(t)$, we consider the case where that the population readout is fast, while the single neuron readout, related to a homeostatic readout of single neuron's firing frequency [10], is a slower process. Such a relation of time constants, $\tau_y < \tau_i^{r,y}$ for $y \in \{E, I\}$, constrains the local current in eq. 17 to spike-triggered adaptation [27]. If, on the contrary, we assume the following relation: $\tau_y < \tau_i^{r,y}$, we get spike-triggered facilitation, a current that might be relevant during learning and restructuring of synaptic connections [39]. Besides the difference of time constants between the population and the single neuron readout, the strength of the local current depends on the regularizer $\mu_y$. In loss functions, the regularizer weights the importance of the metabolic cost over the coding error (eq. (3a) -(3b)). A strong regularizer $\mu_y$ implies that the average firing rate contributes strongly to the loss evaluated by the loss function, a situation where the spiking frequency should be kept in check to keep the loss as small as possible. The weighting of the metabolic cost over the coding error in the loss functions is implemented mechanistically as the local current in eq. 17. Its connection with the loss function makes most sense when the local current is spike-triggered adaptation. An increase of the regularizer $\mu_y$ increases the importance of the metabolic cost in the loss functions, mechanistically increasing the amplitude of adaptation, which reduces the firing frequency of the spiking neuron.

Finally, we derived a depolarizing current triggered by the spike of the local excitatory neuron, that we termed $I_i^{\text{rebound}}(t)$. In the efficient spiking network, the diagonal of the E-to-E recurrent connectivity matrix $D^{EE}$ (eq. 15) implements a positive "self-connection" - as the neuron spikes, it generates a local depolarizing current with slower kinetics of the low-pass filtered spike train. We write this effect as follows:

$$I_i^{\text{rebound}}(t) = \tau_E D_{ii}^{EE} z_i^E(t), \tag{18}$$

with $z_i^E(t)$ the synaptic filter as in eq. 15. Since the matrix $B$ is constrained to be positive semi-definite (see eq. 16), the current in eq. 18 is always depolarizing. After a spike, the local neuron is reset to its reset potential, which, in E neurons, is necessarily below the resting potential (see eq. 11; the resting potential is here at 0 mV). Right after the spike, the local neuron is therefore strongly hyperpolarized, and precisely at this moment, the rebound current activates. The current $I_i^{\text{rebound}}(t)$ thus creates a rebound of the membrane potential after a strong hyperpolarization. In biology, a current with these properties is the hyperpolarization-activated cation current (also called h-current [38]), a current important for the generation of network oscillations, spontaneous firing and in controlling the excitability of cortical neurons [28, 42].

### 2.3 Effect of complexity of E-E connectivity on network responses

In the following, we analyze the effect of the complexity of E-E connections on network responses with analytical considerations and with simulations. We start by considering a simplified network model where we assume the matrix $A$, that determines the transformation between the external input $s(t)$ and the internal signal $x(t)$ (eq. 8), to be a diagonal matrix. A simple way to do so is to set the elements of the matrix $B$ to 0, i.e., $B = (b_{mn})$; $b_{mn} = 0 \; \forall m, n = 1, \ldots, M$, which constrains the desired signal to a leaky integration of input features that is independent across features,

$$\dot{x}(t) = -\lambda_E x(t) + s(t). \tag{19}$$

The time constant of the desired signal $x(t)$ is now equal to the time constant of the population readout of the E cell type in eq. (1), and such a network is devoid of E-E and slower component of E-I (eq. 15) synaptic currents, as well as of the rebound current (eq. 18). We further simplify the network by considering the special case where the time constants of the population and the single neuron readout are equal, $(\tau_i^{r,y}) = \tau_y \; \forall i; y \in \{E, I\}$, which sets the local currents in eq. (17) to zero.

To simulate the network (the simplified one or a more complex one), we have to set values for decoding weights $w_{mi}^y$, which are free parameters. We use unstructured decoding weights that we draw from the normal distribution with zero mean and standard deviation $\sigma_w^y$, $w_{mi}^y \sim \mathcal{N}(0, \sigma_w^y)$, with $\sigma_w^E = 1$ and the ratio $\sigma_w^I : \sigma_w^E = 3 : 1$. We chose decoding weights in the I type to be stronger than in the E cell type, but the number of I neurons is smaller then the number of E neurons, with a biologically plausible ratio $N^E : N^I = 4 : 1$. Decoding weights then determine the strength of recurrent synaptic connections (eqs. 14 and 16).

As we stimulate the simplified network with a step input in one of the input features, the network responds with asynchronous spiking of neurons that are aligned to the active feature (Figure 1B). The feedforward current to neuron $i$ depends on the alignment of the input $s(t)$ with neuron's decoding vector $w_i^E$ (see eq. 12). When a single feature is activated, e.g. $s_n(t) \neq 0$, a positive input, $s_n(t) > 0$, drives neurons with positive decoding weight for the $n-$th feature (i.e., neurons with $w_{ni}^E > 0$), while a negative input, $s_n(t) < 0$, drives neurons with negative decoding weights (neurons with $w_{ni}^E < 0$). If several features are active simultaneously, the feedforward current to E neurons is given by a linear mixture of features. The simplified network is purely feedforward-driven and the network response never outlasts the stimulus (Figure 1B). The recurrent connectivity of such a network consists of structured fast connections between neurons with similar decoding selectivity (eq. 14), and the stronger the similarity of decoding vectors of neurons $i$ and $j$, the stronger their synapse.

We then considered a network with E-E and slower E-I connections. In general, E-E and slower E-I connections emerge when the matrix $A$ in eq 8 is non-diagonal. The minimal requirement for the emergence of E-E and slower E-I connections is that at least one diagonal element of the matrix $A$ in eq. 8 is different than the inverse membrane time constant of E neurons, $\lambda_E$. Recurrent excitatory (E-E) connections and slower E-I connections, $D^{EE}$ and $D^{IE}$, are then determined by the similarity of decoding vectors $w_i^E$ and $w_j^E$ (same as fast connections), as well as on the matrix $B$ (in contrast to fast connections; see eqs. 14 and 16). Square matrix $B$ is constrained to be positive semi-definite to ensure that the network obeys Dale's law (eq.16), and positive semi-definiteness in turn constrains the matrix $B$ to be symmetric and to have non-negative eigenvalues. To ensure that these properties are satisfied, we write the matrix $B$ as follows:

$$B = a\Gamma\Gamma^{\mathsf{T}}, \qquad \Gamma = [b_1, \ldots, b_{M'}] \tag{20}$$

with $a > 0$ a positive constant influencing the strength of E-E and slower E-I connections, and $b_1, \ldots, b_{M'}$, a set of linearly independent column vectors, with entries

$\boldsymbol{b_n} = [b_{n1}, \ldots, b_{nM}]^\mathsf{T}, n = 1, \ldots, M'$, and where the number of vectors is constrained as follows: $1 \leq M' \leq M$. In it important to consider that matrix $\Gamma$ determines the rank of the matrix $B$, as $M'$ linearly independent column vectors set the rank of $B$ to rank$(B) = M'$. The rank of the matrix B in turn determines the rank of connectivity matrices $D^{EE}$ and $D^{IE}$ through eq. (16).

The response of the network now critically depends on the complexity of recurrent excitatory (E-E) connections. In the simplest case, beyond the trivial case where all elements of $B$ are set to zero that we discussed above, $\Gamma$ is defined with a single column vector, $\Gamma = \boldsymbol{b_1}$. This sets the matrix $B$ to the following: $B = a\boldsymbol{b_1}\boldsymbol{b_1}^\mathsf{T}$, and determines the rank of the matrix $B$, as well as the rank of the E-E connectivity matrix $D^{EE}$, to rank$(B) = $ rank$(D^{EE}) = 1$. We simulate such network, setting the membrane time constants of E and I cell type to biologically plausible values, with $\tau_E = 10$ ms for pyramidal neurons, and $\tau_I = 5$ ms for inhibitory interneurons, as measured empirically in the cerebral cortex [47]. In addition, we assumed time constants of the single neuron readout to be longer than the time constants of the population readout, as single-neuron readout is assumed to be a slower process of homeostatic regulation of the firing rate in single neurons. In particular, we set the time constant of the single neuron read-out to be twice as long as the time constant of the population read-out. According to eq. (17), such relation of time constants between the single neuron and population read-out gives spike-triggered adaptation in E and I neurons.

As we simulated such a network, we found a population-wide oscillation in the network activity (Figure 1C). Such a population-wide oscillation is in contrast to the response of the simpler network without E-E connections and without spike-triggered adaptation, where the population firing rate with constant stimulus converges to a constant value after an initial transient (Figure 1B). Similarly to the simpler network, however, we still have that only neurons aligned with the input feature are active (e.g., neurons with $w_{1i}^y > 0$ when the active stimulus feature is $s_1(t) > 0$).

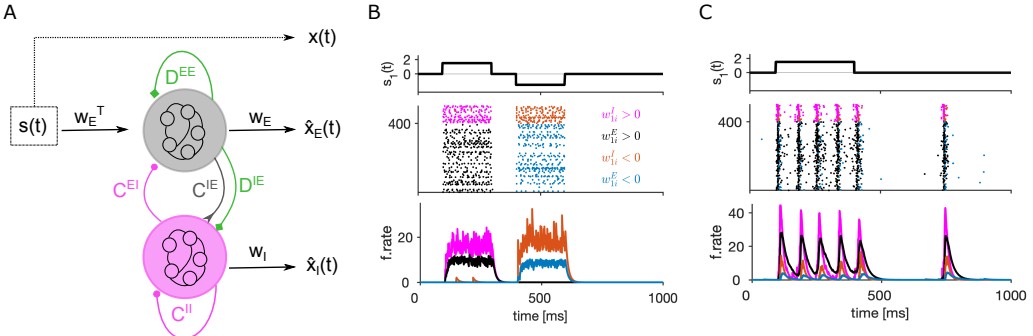

Figure 1: (A) Schema of the E-I network. E neurons (gray) receive a linear sum of external inputs $\boldsymbol{s}(t)$. E and I (magenta) neurons with similar selectivity are recurrently connected. Inhibitory synapses $C^{EI}$ and $C^{II}$ have fast kinetics, E-to-I synapses have a fast and a slower component ($C^{IE}$ and $D^{IE}$), while E-to-E synapses only have the slower kinetics $C^{II}$ and , as well as through slower synapses $D^{EE}$ and (green). Linear population readout of the spiking activity of E and I cell types define estimates $\hat{\boldsymbol{x}}_E(t)$ and $\hat{\boldsymbol{x}}_I(t)$, respectively. (B) Activity of the simplified network without E-E connections and without the adaptation current. Top: Step input in the first feature $s_1(t)$. The remaining features are inactive. Middle: Spike trains of E and I neurons aligned with the positive input in the first feature (black and magenta, respectively), and aligned with the negative input in the first feature (blue and red, respectively). Bottom: Neuron-averaged firing rate, with the same color code as in the middle plot. Parameters: $M = 3$, $N_E = 400$, $N_E : N_I = 4 : 1$, $\tau_E = \tau_I = \tau_i^{r,E} = \tau_i^{r,I} = 10$ ms $\forall i$, $b_{mn} = 0 \,\forall m, n = 1, \ldots, M$, $\mu_E = \mu_I = 6$, $\sigma_i^E = \sigma_i^I = 0.167 \,\forall i$, $\sigma_w^E = 1$, $\sigma_w^I : \sigma_w^E = 3 : 1$, $dt = 0.02 \, ms$. (C) Same as in B but showing the activity of the network with E-E connections and where E-I synapses have both fast and slow component. The network also has adaptation current in both E and I neurons. Parameters: $\tau_E = 10$ ms, $\tau_I = 5$ ms, $\tau_i^{r,E} = 20$ ms $\forall i$, $\tau_i^{r,I} = 10$ ms $\forall i$, $B = a\boldsymbol{b_1}^\mathsf{T}\boldsymbol{b_1}$ with $a = 0.035$ and $\boldsymbol{b_1} = [0.5, 0, 0.3]^\mathsf{T}$, $\sigma_i^E = \sigma_i^I = 0.25 \,\forall i$. Other parameters as in B.

With higher rank of E-E connectivity, the network response reflects mixing of input features. We now define the matrix $B$ with $M' > 1$ linearly independent vectors, where $a_1\boldsymbol{b_1} + \cdots + a_{M'}\boldsymbol{b_{M'}} = \boldsymbol{0}$ if and only if $a_n = 0 \,\forall n$. Linear independence of column vectors $\boldsymbol{b_n}$ implies that the matrix $B$ has rank $M' > 1$. The rank of the matrix $B$ than determines the rank of

E-E connectivity matrix due to eq. (16). We find that with rank of E-E connectivity bigger than 1, also neurons that are not aligned with the active input feature participate in the network response. As shown on Figure 2A-B, with $s_1(t) > 0$, excitatory neurons with a positive decoding weight for the first feature, $w_{1i}^E > 0$ (black), have similar firing rate than neurons with negative decoding weight for the same feature, $w_{1i}^E < 0$ (blue). Since the feedforward current only drives neurons with decoding weights aligned to the input feature (eq. 12; here neurons with $w_{1i}^E > 0$ in black), this means that the response of misaligned neurons (i.e. neurons with $w_{1i}^E > 0$), is driven by the E-E connectivity. In summary, E-E connectivity with rank 1 only drives neurons that are aligned with the active stimulus feature, while higher rank of E-E connectivity also drives misaligned neurons.

For a fixed strength of E-E connectivity, increasing the noise results in longer duration of network's response to the stimulus, and in spontaneous activation of the network long after the external input is turned off (Figure 2B). In the absence of the external stimulus, such a network responds with rhythmic spontaneous activation (Up state), followed by a period of silence (Down state; Figure 2C). Up states are triggered by the noise at threshold, that by itself occasionally leads to a spike of a single neuron (see eq. 10). A random spike is amplified by the recurrent excitatory synaptic activity, and spontaneous activity is suppressed by the adaptation current.

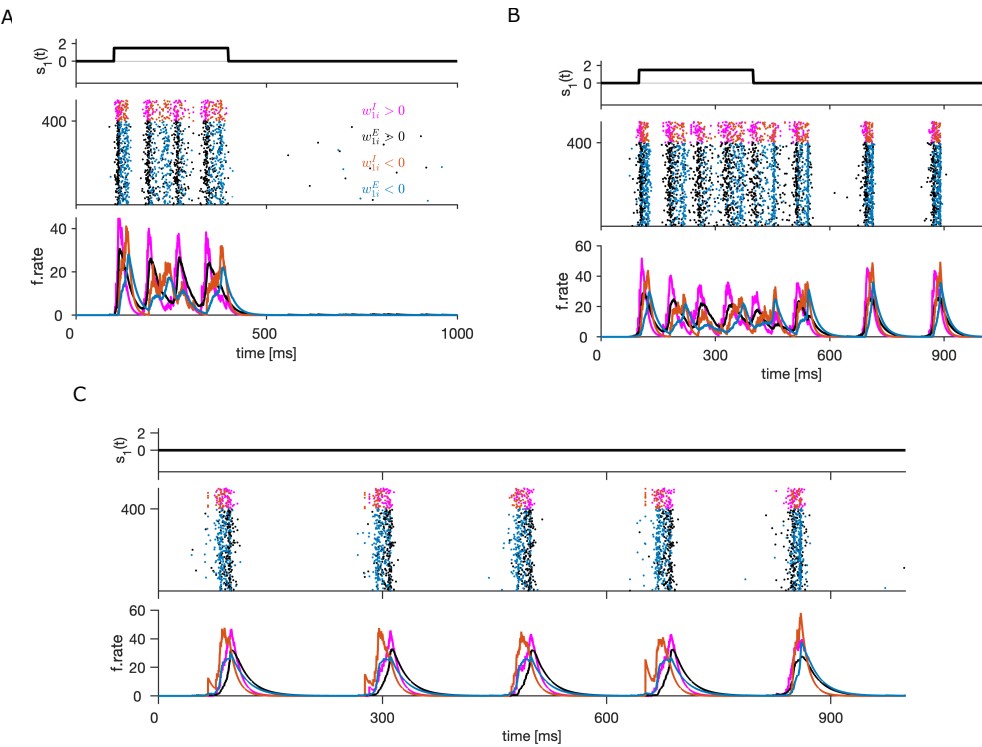

Figure 2: (A) Same as in Figure 1C, but using the matrix $B$ with rank bigger than 1. To construct the matrix $B$, we impose the rank $M' = M = 3$ and use the following column vectors: $b_1 = [0.4, 0.1, 0.1]^\mathsf{T}$, $b_2 = [0.1, 0.4, 0.1]^\mathsf{T}$, $b_3 = [0.1, 0.1, -0.4]^\mathsf{T}$. Other parameters as in Figure 1C. (B) As in A, but with increased strength of the noise. Parameters: $\sigma_i^E = \sigma_i^I = 0.3 \; \forall i$. Other parameters as in A. (C) Same as in B, but without the external stimulus, $s_m(t) = 0 \; \forall m, t$. In the network with rank of E-E connectivity matrix larger than 1, all neurons participate in the population responses.

## 3 Discussion

We developed an E-I spiking network that makes information coding efficient under biologically relevant constraints. Our model is made efficient by minimizing two loss functions comprising a quadratic estimation error and a quadratic regularizer, and assuming an arbitrary linear input/output transformation. The general solution of an online minimization of loss functions with spikes is a

generalized LIF model with a specific set of structural and dynamical features. Structural features include low-rank structured connectivity [46, 22, 24], where fast synapses between E and I neurons implement competition between neurons with similar decoding selectivity and slower synapses implement cooperation. Moreover, structural features resulting from optimization with spikes are external inputs that are shared across neurons, spike-triggered adaptation currents in single neurons and hyperpolarization-triggered rebound current in E neurons. Assuming heterogeneous decoding weights, all currents with the exception of leak currents are heterogeneous across neurons, increasing the biological plausibility of the model.

Similarly to the seminal paper by Boerlin and colleagues [3], the spiking network proposed here assumes that spikes are fired if and only if they minimize the quadratic error between the signal and the population readout. As the network model is analytically developed from such optimization principles, efficient coding is a build-in property of the network. In earlier works on efficient coding [5], however, the leak current is not derived exactly but only added to the membrane equation for biological plausibility. Our derivations define an exact expression for the leak current, similarly as in [23]. Contrary to most previous works on efficient coding with spikes that use a single loss function to derive the membrane equations, we here propose two separate loss functions that relate to the activity of E and I neurons. With this, we impose the E-I architecture and allow for functional diversity between E and I neurons. A previous proposal that already used separate loss functions for E and I neurons [7] yielded a network similar to our simplified network, but the behavior or their network is markedly different than our simplified model, possibly due to very long and unrealistic synaptic time constants they use.

Most previous proposals on efficient coding [6, 7, 23, 5, 2] assumed a specific neural computation, namely a leaky integration of external inputs (but see [3, 1]). We here developed the case of the general linear transformation between the external inputs and the internal signals of the network with a biologically plausible spiking network. This generalization allowed us to formulate a biologically plausible expression for E-E connections and to gain insights on how the structure of the mixing matrix qualitatively changes network responses. Without mixing of features, only neurons that are aligned to the external stimulus, and consequently driven by the feedforward current, participate in network responses. With increased complexity of E-E connections, we also observe activation of neurons that are not driven by the feedforward current, but are instead engaged through recurrent E-E connections. Moreover, in the absence of the external stimulus and with sufficient background noise, such a complex network produces spontaneous Up and Down states that could describe spontaneous Up and Down states observed in biological networks [45].

Our results demonstrate that the minimization of the quadratic loss function with spikes results in a generalized LIF model, a spiking neuron model that has been shown to provide a good fit to the spiking dynamics of biological networks [14, 17]. An important advantage of the present computational framework over previously proposed generalized LIF models is the access to a functional interpretation of network parameters. In general, a change in a network parameter can have a strong influence on network's dynamics. Besides characterizing the effect on neural dynamics, it is insightful to also have the interpretation of the role of a specific parameter on the functional/computational level. In the present setting, parameters of the spiking network are directly related to the variables minimized in loss functions, which allows to connect their effect on dynamics with their role for network's computation.

In the present work, we emphasized the theoretical development of an optimization-based approach towards a biologically plausible spiking neural network with complex, non-linear responses. Future work could introduce performance measures, for example by measuring the distance between the signal and the population readout, or by using information-theoretic measures [4, 31, 41]. Moreover, future work could perform an analysis of network parameters and assess how are particular dynamical phenomena, such as rhythmic spontaneous Up and Down states [45], related to information processing in efficient spiking networks.

## Acknowledgments

This work was supported by the NIH Brain Initiative Grants U19 NS107464, R01 NS109961, and R01 NS108410.

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
