# OpenReview forum: "Biologically plausible solutions for spiking networks with efficient coding"
_NeurIPS.cc/2022/Conference — NeurIPS 2022 Accept_

### Official Review · Reviewer_DCGS · 2022-07-07

**Rating:** 6
**Confidence:** 3
**Soundness:** 3 good
**Presentation:** 2 fair
**Contribution:** 3 good

**Summary:**

The authors present a spiking network model with E/I neurons along with two loss functions which include a squared loss to some stimulus and a quadratic regularizer. From here they show how this model matches a generalized leaky-integrate-and-fire (LIF) network under certain approximations. In this sense the LIF network is ‘optimal’ in the given framework. The authors finally show different network dynamics, first for a simplified network without E-E connections, and later for the full network with varying ranks of the connectivity matrix. The authors highlight their ability to interpret and link different model components to biological processes.

**Questions:**

- Why are the loss functions meaningful? While 3a makes sense (if the readout should replicate the signal), 3b is less intuitive. The authors could give some more explanations and context.

**Limitations:**

The authors mention their limited evaluation and list further directions for analysis and investigations. They do not comment on negative societal impacts.


**Strengths And Weaknesses:**

### Strengths
The authors give an interesting link of generalized LIF networks to the optimization of specific loss functions. The derivations are extensive and technical sound. The presented model has rich properties and can be used for further analysis. It has the potential to link existing spiking network models with more biological interpretable models and model components.

However, there are several weaknesses:

### Weaknesses
1. While ‘efficient coding’ is one of the main selling points of the authors, it is in the end an optimization of quadratic error term with a quadratic regularizer. As the exact formulation of efficient coding and the made assumption play a major role in efficient coding, the authors should give more context and state clearly the made assumptions.
2. The main sections 2.1 and 2.2 are written quite technical and miss some clarity. A clear road map could help to guide through the technicalities.
3. The made assumptions and approximations as well as the derivation from (6) to (9) are hidden and it is hard to grasp their correspondence.
4. The analysis of the presented network is limited and an interpretation of the results is missing. What is for example the purpose of investigating the simplified network? Why is this interesting? What are the properties of the network? What influence have the different model parameters? What influence has the noise level?
5. As the whole setup is formulated as an ‘efficient coding problem’, it would be interesting, if not essential, to see the performance of the network for different configurations and/or stimuli.

### Minor comments:
- Figures are rather small, and could be at least shown with full text width.

---

> ### Author Response · Authors · 2022-08-01
> **Interesting and valid review**
>
> Strengths and weaknesses
>
> 1) We clarified the concept of efficient coding in our revised paper.
> We used the term “efficient coding” since this terminology was used in previous related works (Alemi et al. 2018, Brendel et al. 2020). The term “efficient coding” refers to the concept where neurons fire spikes only when their firing minimizes the error between the desired signal and its estimate (given by the population read-out). This concept is formalized with loss functions that include a quadratic error between the desired signal and the estimates. A spike is fired if and only if it minimizes the loss function, making each spike informative for the coding task. Certainly, not all parameter setting will result in accurate coding in the actual spiking network, however, as the network is derived from the loss functions, “efficient coding” is a build-in property of the network. A more descriptive term for our approach could be “error-correcting coding with spikes”, a term we would like to consider in our revised paper in case the reviewer finds it more appropriate.
>
> 2) We provided a road map for technical sections 2.1 and 2.2.
>
> 3) We clarified the assumptions and the analytical procedure between eq. 6 and 9. Extensive details are provided in the Supplementary material, as the detailed description is too long to be exposed in the main part of the paper.
>
> 4) We agree that the analysis of the network is limited and we are running further analysis to provide more results. Due to space limitations, we focused on what we estimated to be the most important aspects of the network to analyze, namely, how the changes in E-E connectivity cause qualitative changes in network dynamics. Following your hint, we removed the section on the simplified network.
>
> The noise level in simulations was set to a subthreshold value, with the exception of the last plot (spontaneous Up and Down states) where the noise is sufficiently strong to trigger an Up state (but the rest of the spiking is due to the network). Previous work on efficient coding with spikes (Koren & Deneve 2017) has found that there is an optimal level of noise for network performance, however, they used a substantially different network of lesser biological plausibility. We are analyzing the effect of other network configurations and different types of stimuli on network dynamics.
>
> 5) We are also analyzing different network and stimulus configurations as a function of the performance of the network.
>
>
> Reply to Minor comments: We made the figures full text width.
>
> Answer to the Question
> This is a pertinent question. We decided for such a formulation because it allows to derive fast connections between E and I neurons that are important for efficient coding. Structured fast connections between E and I neurons, as they follow from the loss function as in eq. 3b, underlie lateral inhibition between E neurons with similar selectivity. This effect is crucial for coding efficiency, since it prevents redundant spiking. In particular, many E neurons with similar selectivity are depolarized by the shared external input, and lateral inhibition makes sure that only the necessary number of spikes are fired to correct the error between the signal and the E estimate.
>
> If I neurons minimize the distance between the desired signal and the population read-out of I neurons, our preliminary analysis gives an E-I network where E and I populations are not connected with fast connections, and unconnected E and I networks defy the purpose of the E-I architecture. Possible further approximations could allow to use the objective function where I neurons minimize the distance between the signal and the I estimate, however, we find that these approximations are beyond the scope of the present paper.
>
> Another reason for the proposed formulation of the objective function of I neurons is to naturally put in place a time-dependent balance of E and I currents in single neurons.  With the proposed formulation as in eq. 3b, the E and I estimates track each other at every time step. The consequence is the balance of E and I currents in single neurons that occurs not only on average, but also over time, in the sense that we get E and I currents that are strongly temporally correlated, similarly to what has been demonstrated in experiments (Okun and Lampl, 2008).
>
> Reply to Limitations: We find there is no negative societal impact that could arise from the paper.

---

> > ### Comment · Reviewer_DCGS · 2022-08-05
> > **Re**
> >
> > I thank the authors for the detailed response and the clarifications.
> >
> > It is a bit unfortunate that the authors did not update their manuscript such that the promised changes can be evaluated.
> >
> > As some of my major concerns were about clarity, I will leave my score as I can not judge the updates on the manuscript.

---

> > > ### Author Response · Authors · 2022-08-05
> > > **response to reviewer**
> > >
> > > Thank you. We are working on the revision right now and will submit it in less than 24h from now, hopefully the reviewer can still consider the revised paper.

---

> > > ### Author Response · Authors · 2022-08-06
> > > **Revised paper**
> > >
> > > We thank the reviewer for their time and contribution. We uploaded the current revisions we made in response to the Reviewer’s suggestions. Parts of the paper that were revised the most are highlighted in blue font. These were the main changes we implemented following the Reviewer’s suggestions. Any feedback that they may have would be much appreciated. We made the following main changes:
> > >
> > > •	we better clarified the concept of efficient coding that we specifically used in this work-see Introduction (lines 30-31 and 53-54) and Discussion (lines 324-325)
> > > •	we added a “road map” through technical sections (last paragraph of Introduction; lines 74-80)
> > > •	we provided a justification of the shape of the loss function of I neurons (lines 105-107)
> > > •	we clarified the summary and assumptions of the analytical procedure between eqs. 6 and 9 (line 137-138)¬
> > > •	we clarified the description of the rebound current (lines 235-237)
> > > •	we removed the section on the simplified network (the content of the section is now summarized in the first paragraph of the section 2.3 (lines 242-249) and on Figure 1b
> > > •	we added a paragraph in Discussion to relate current model to related works (lines 334-345)
> > >
> > > Although we do not have sufficient space in the paper (and sufficient time within the openreview timeframe) to fully extend the network in many possible directions and fully explore the parameter landscape, we extended the study with the analyses that were, in our view, of biggest interest. In particular, we analyzed how an increase in complexity of E-E connections (lines 287-298, Fig 1A and Fig 2) causes qualitative changes in network responses. The synaptic weight between E neurons is determined by the similarity of decoding vectors and the matrix B (eq.18), and the matrix B is defined from the feature matrix A (eq. 8). If the matrix B is determined by a single column vector, only neurons whose decoding vector is aligned with the input feature respond (Figure 1C), as they are driven by the feedforward current. A higher rank of the matrix B in addition causes a response in neurons that are not aligned with the input feature. This is interesting because with the rank of B higher that 1, encoding of a single input feature is now distributed among a larger pool of neurons and the decoding weight of the neuron can no longer predict if the neuron will respond or not. The inclusion of E-E connections revealed network responses with highly non-linear dynamics, driven by feedforward currents as well as by recurrent network connections.

---

> > > > ### Comment · Reviewer_DCGS · 2022-08-08
> > > > **Re: revised manuscript**
> > > >
> > > > I am not sure if you are still allowed to upload a revised manuscript after the rebuttal deadline.
> > > >
> > > > I can only say, that I am not able to download the newest version of your work.

---

> > > > > ### Author Response · Authors · 2022-08-08
> > > > > **response**
> > > > >
> > > > > We apologise, it seems that something went wrong with the upload. We uploaded it again, the pdf is now visible for us on the openreview. We hope it is not too much of an inconvenience.

---

### Official Review · Reviewer_QSDp · 2022-07-11

**Rating:** 6
**Confidence:** 3
**Soundness:** 3 good
**Presentation:** 3 good
**Contribution:** 2 fair

**Summary:**

The authors derive a neural network model that optimizes the information transfer. The optimization leads to a number of biologically relevant features such as generalized leaky integrate-and-fire neuronal dynamics, low-rank connectivity and heterogeneous decoding weights and currents across neurons in the network.

**Questions:**

I would like to ask the authors to at least propose experiments that could validate their results.

**Limitations:**

Yes.

**Strengths And Weaknesses:**

In terms of originality, this work continues a line of research that builds biologically plausible neural networks whose objective is to maximize information coding. In that line of research, this works presents a leap forward in the right direction because it adds a number of biologically realistic features. It is my understanding that this works follows previous work closely so I would argue that the novelty is limited. The paper itself seems solid and there is no major errors I can see in the derivations. The paper is written in a clear enough manner although a section that explains the structure of the paper and/or a section that clearly outlines the contributions of this work would make it easier to read.  In terms of significance, I would argue that it is hard to judge because it is generally hard or impossible to validate how biologically-plausible these features are. It is without a doubt that features such as heterogeneity are biologically plausible, but I miss a way to validate this claims more quantitively. Also it is not completely clear to me from the paper how these different features are contributing to make the coding more efficient.

---

> ### Author Response · Authors · 2022-08-01
> **Valid and insightful review that can guide the revision**
>
> Following reviewer’s suggestion, we added in our revised paper a section that outlines the structure of the paper. We clarified the contribution of this work to the field and also added a paragraph in discussion where we comment on the relation of our model with related works.
>
> As of biological plausibility of structural features derived in our paper, we would argue these are of substantial biological relevance. Mentioned properties have many times been reported in experiments on biological brains in-vivo and in-vitro and we argue there is little doubt about their biological plausibility. Some of these properties, such as spike-triggered adaptation, heterogeneity of decoding weights, connectivity weights and currents, is textbook knowledge in neuroscience (see e.g., Dayan and Abbott, 2003; Gerstner et al. 2014). Other properties such as low-rank connectivity and complexity of network responses, are suggested by the theory (Sompolinsky et al. 1988) and by experimental studies (Tseng et al. 2022), but remain open questions since they are difficult to measure.
>
> We agree that an interesting question is if these properties make coding more or less efficient. In our preliminary analysis we found that the “simplified network” without E-E connections and without adaptation estimates the desired signal very precisely with the read-out of E as well as I neurons. Additional biologically plausible features such as adaptation and E-E connections do not seem to improve the accuracy of the model, at least when where the performance is measured as the coding accuracy-the average error between the signal and the E estimate. However, besides the coding accuracy, the objective function (eq. 3) also comprises the regularizer that depends on instantaneous firing rates r_i(t), motivated with the suggestion that biological systems care about the metabolic cost on spiking (Kuzawa 2014) and not only about coding precision. We reason that a possible metric of network performance could therefore take into account also the firing rates along with the coding error. The adaptation in E neurons decreases firing rates and can increase the performance of the network if the performance measure includes metabolic cost.  We are currently working on this question and would consider adding a result in the revised paper if the reviewer finds it essential.
>
> We propose two ways to test the model. One is by simulating the network of a realistic size and utilizing known biophysical parameters for time constants, adaptation strengths, synaptic (connectivity) weights etc. Does such a model reproduce dynamics that is typically observed in biological networks, including state-dependent activity? In the present simulations, we used realistic numbers for membrane time constants of E and I neurons and their relation, for synaptic time constants, synaptic weights and the ratio of E-to-I neurons. However, a more systematic approach could include bigger networks, and be complemented by measuring network performance.
>
> The second way to test the model could be to fit the model and to the spiking activity of cortical networks in-vivo. We suggest to use recordings of parallel spike trains from an experiment on behaving animals where we can infer the computation performed by a given biological network (e.g., the primary visual cortex). From such spiking data, we can determine dependent variables that the network might be computing (e.g., the orientation of the visual stimulus), estimate decoding weights (e.g., with a linear classifier such as the Support Vector Machine), as well as the rank of the connectivity using dimensionality reduction methods (with PCA or related methods). With estimated parameters from the data, we could constrain the free parameters of the model network, and see to what extent the spiking dynamics of the held-out data is captured by the model.
>
> Our model also has several specific predictions that could soon be tested on data using new experimental methods. For example, our model predicts moderate to strong correlation of E and I current in single neurons over time. Simultaneous monitoring of E and I current in the same neuron is still a challenge, but might soon be possible to realize with new recording techniques (Müller-Komorowska et al. 2021). Moreover, our model predicts particular connectivity structures where neurons with similar decoding selectivity are more strongly connected than neurons with different selectivity (eq. 15). New recording techniques in electrophysiology allow to estimate the functional influence between local neurons in-vivo (Chettih and Harvey, 2019), raising the possibility of measuring the connectivity structure in local networks and to test the prediction of precise connectivity structure predicted by the theory.
>
> We will include some of this discussion in the revised paper.

---

> > ### Comment · Reviewer_QSDp · 2022-08-08
> > **No change to my review needed**
> >
> > I am happy with the author's response. I think they have made an effort to address all my comments. Given the authors response and the other reviews, I see no reason to change my ratings. I still think this paper could be accepted in the conference.

---

### Official Review · Reviewer_SvZ9 · 2022-07-12

**Rating:** 2
**Confidence:** 5
**Soundness:** 1 poor
**Presentation:** 1 poor
**Contribution:** 1 poor

**Summary:**

As I think the authors do not respect the NeurIPS reviewers, I just copied what they write in the abstract.

"we find a biologically plausible spiking model realizing efficient coding in the case of a generalized leaky integrate-and-fire network with excitatory and inhibitory units, equipped with fast and slow synaptic currents, local homeostatic currents such as spike-triggered adaptation, hyperpolarization-activated rebound current, heterogeneous firing thresholds and resets, heterogeneous postsynaptic potentials, and structured, low-rank connectivity."

**Questions:**

Typos:

1. line 12: remove the first "and"
2. line 340: remove the first "how"
3. line 79: "...of the cell type Y and M ...", missing a "," before "and"




**Limitations:**

No insights and no background. The authors even did not change the format from the previous submission.

No experimental evidence on the application level supports the proposed method. In addition, I did not see the bio-plausible support either.

**Strengths And Weaknesses:**

Weaknesses:

-1. The authors should reformat the manuscript to NeurIPS format, i.e., introduction, related work, method, results, discussion & conclusion. If the manuscript was for nature, science, etc., please reformat it! It is not respectful for NeurIPS reviewers!

-2. The authors set the objective of the inhibitory population to minimize the distance between E and I. But, no explanation about the objective. Why is it? Then, we can set the objective of the excitatory network to minimize the distance between the desired signal and I?

-3. lines 109-111: it is not accurate. Not only $Na^+$ and $K^+$ are responsible for the potential changes. Which biological model are the authors talking about?

-4. lines 126-128: It is unclear how Eq. (8) can be re-arranged to the LIF model.

-5. I did not see any experimental evidence on the application level to support the effectiveness of the proposed approach.

In summary, the manuscript lacks insights. The authors only describe the mathematical models without providing any insights and background knowledge. The writing style is poor.

---

> ### Author Response · Authors · 2022-08-02
> **Inappropriate review**
>
> Dear reviewer,
>
> This is a first submission and has been though specifically for Neurips. We estimated that the analytical work behind our results is important to present as thoroughly as possible, and have decided to submit our paper to Neurips because the format of the conference paper at Neurips allows a more thorough presentation of the analytical work than most neuroscience journals and conferences. Moreover, the methods we present in our paper come from fields of both computational neuroscience and machine learning, which is fitting with the scope of Neurips.
>
> The reviewer asks for the justification of the objective function of I neurons.
> While the objective function of E neurons minimizes the distance between the desired signal and the population read-out of E neurons (eq. 3a), the objective function of I neurons minimizes the distance between the population read-out of E and I neurons (eq. 3b). We decided for such a formulation because it allows to derive fast connections between E and I neurons that are important for efficient coding. Structured fast connections between E and I neurons, as they follow from the loss function as in eq. 3b, underlie lateral inhibition between E neurons with similar selectivity. This effect is crucial for coding efficiency, since it prevents redundant spiking. In particular, many E neurons with similar selectivity are depolarized by the shared external input, and lateral inhibition makes sure that only the necessary number of spikes are fired to correct the error between the signal and the E estimate.
>
> If I neurons minimize the distance between the desired signal and the population read-out of I neurons, our preliminary analysis gives an E-I network where E and I populations are not connected with fast connections, and unconnected E and I networks defy the purpose of the E-I architecture. Possible further approximations could allow to use the objective function where I neurons minimize the distance between the signal and the I estimate, however, we find that these approximations are beyond the scope of the present paper.
>
> Another reason for the proposed formulation of the objective function of I neurons is to naturally put in place a time-dependent balance of E and I currents in single neurons.  With the proposed formulation as in eq. 3b, the E and I estimates track each other at every time step. The consequence is the balance of E and I currents in single neurons that occurs not only on average, but also over time, in the sense that we get E and I currents that are strongly temporally correlated, similarly to what has been demonstrated in experiments (Okun and Lampl, 2008). A justification of the loss function has been aded to the revision.
>
> Reviewer mentions that not only sodium and potassium are responsible for the potential changes. Certainly, several other ions are in general involved in potential changes, however, what we do is to provide an interpretation of an analytically derived current that is the most plausible in the setting of biological neurons. Our analytical derivations give a depolarizing current that activates when the single neuron is strongly hyperpolarized. The effect of this current is to create a rebound of the membrane potential towards the resting potential after strong hyperpolarization. We found that the biophysical current that corresponds best to such effect is the h-current, a current that has been observed experimentally in biological neurons. The abstract nature of our mathematical derivation does not aim to describe the h-current in all its biological details, but only captures the most important effect on neural dynamics (i.e., the rebound of the current upon hyperpolarization). We also do not aim to describe all possible effects on the potential changes that have been reported in experimental literature. We simply interpret a mathematically derived current. We clarify this aspect in the revised paper.
>
> Reviewer mentions it is unclear how eq. 8 can be arranged to LIF model. Most likely the reviewer refers to eq. 9a-11b, with currents specified by eq. 12-19? These define a generalized LIF model according to generally accepted terminology (see for example Ostojic & Brunel 2011 for the LIF; Gerstner et al. 2014 for the generalized LIF). Also, previous proposals on efficient coding found their model is a LIF model (see for example Boerlin et al. 2013).
>
> Our paper has the scope of providing a theoretical framework for coding and dynamics in recurrently connected neural networks in biological brains. Similarly to many other influential papers, we do not analyse the data or provide experimental evidence, but provide a rigorous analytical development of a network that is generally accepted as a biologically plausible model.
>
> The simulation of the network reveals dynamical features that are omnipresent in biological networks, but not yet well understood. Our theory gives an opportunity to understand these phenomena.

---

> > ### Comment · Reviewer_SvZ9 · 2022-08-02
> > **Response to the authors**
> >
> > Thanks for the feedback! But, to be fair, I received the rebuttal after the rebuttal deadline. To be more precise, 2 hours 5 mins after the deadline. Therefore, I regarded it as no response!
> >
> > "Inappropriate" is very subjective! I believe the questions I asked are valid and respectful. It is good that it is an open review. All researchers could see the reviews.
> >
> > I have already brought my concerns to the Chairs. I will not spend any time on this submission, especially after being called an "inappropriate" reviewer.
> >
> > Good luck!

---

> > > ### Author Response · Authors · 2022-08-03
> > > **Response to the reviewer**
> > >
> > > We thank the reviewer for their time.  We have submitted the response to your review to the Chair before the deadline. We then got suggested to send a response to you as well. We did not know that the two types of response can be posted in parallel.
> > >
> > > In the first response to your review, we tried to concentrate on comments that were possible to discuss, but here we comment more broadly on your review.
> > >
> > > The review contains a false accusation (about being disrespectful and submitting to Neurips a paper we did not reformat after a previous submission) and makes the following statements about the paper:
> > >
> > > 1) that the writing style is poor
> > > 2) there are major technical flaws
> > > 3) there is limited impact, and
> > > 4) there is poor reproducibility
> > >
> > > These are very negative statements and the reviewer does not bring any rational arguments, details, explanations, nor examples to justify these statements.
> > >
> > > For example, why does the reviewer accuse our work of poor reproducibility? In the main paper and in the Supplementary material, we have exposed what we believed was all the necessary equations and described all the assumptions we have made. We have submitted a complete code that we used for generating results and figures (including even the generation of plots). What more could be done to ensure reproducilibility? If there is an equation or the line of code that seems unclear to the reviewer or if they see a mistake, or any kind of flaw, they are invited to point it out. However, simply making the statement that the reproducibility is poor is unfortunately not helpful for anyone. Same goes for all other negative statements.
> > >
> > > Moreover, the reviewer gives the score of 1 for Soundness, Presentation and Contribution, again without justification.
> > >
> > > For these reasons, and with all due respect, we found that the review is inappropriate, as it does not bring any justification to very negative statements. It goes without saying that we worked hard with the goal of making a good-quality, rigorous, clear and relevant contribution to the field. If there are no rational arguments, there is no way for us to defend our work, or to improve it, and this defies the scope of a review.

---

> > > > ### Comment · Reviewer_SvZ9 · 2022-08-03
> > > > **Deadline is Deadline**
> > > >
> > > > Valid discussions are based on a valid response, meaning a response before the rebuttal deadline. The authors' response was 2 hours 5 mins late, as evidenced by the email timestamp. To enforce fairness, no one should care about the reasons for missing the deadline.
> > > >
> > > > As promised, I will not spend any time on this submission, meaning no feedback.
> > > >
> > > > The good news is I explicitly asked AC to assign an "appropriate" reviewer to this submission before the authors' first invalid response. So, good luck!
> > > >
> > > > SvZ9, oh, sorry should be "inappropriate" SvZ9 :)

---

> > > > > ### Author Response · Authors · 2022-08-03
> > > > > **Thanks**
> > > > >
> > > > > Dear Reviewer SvZ9,
> > > > > We appreciate your time and work on the paper. Our reply used the term “inappropriate” with reference to the review, not to the Reviewer. We never expressed (and never intended to express) a judgement on the Reviewer.  Thanks again.

---

### Meta-Review · Area_Chair_XTCE · 2022-08-26

**Recommendation:** Accept
**Confidence:** Certain

**Metareview:**

This paper proposes a derivation of spiking networks from an efficient signal reconstruction cost function. The derivation leads to more biological features than previous ones, including fast and slow synaptic currents, and rebound currents. This is a nice addition to the growing literature in this field.

The negative review below mostly focused on formatting of the paper. The AC did not see a violation of NeurIPS formatting policy, however agrees that the presentation could be clearer.

**Award:**

No

---

### Decision · Program_Chairs · 2022-09-14

Accept